# Characteristics of Synaptic Function of Mesoporous Silica–Titania and Mesoporous Titania Lateral Electrode Devices

**DOI:** 10.3390/nano13111734

**Published:** 2023-05-25

**Authors:** Dhanashri Vitthal Desai, Jongmin Yang, Hyun Ho Lee

**Affiliations:** Department of Chemical Engineering, Myongji University, Yongin-Si 17058, Gyeonggi-Do, Republic of Korea; dhanashri.nanotech@gmail.com (D.V.D.); yangjm94@gmail.com (J.Y.)

**Keywords:** mesoporous silica titania, synaptic device, rectification, neuromorphic electronics

## Abstract

In this paper, we have fabricated non-volatile memory resistive switching (RS) devices and analyzed analog memristive characteristics using lateral electrodes with mesoporous silica–titania (meso-ST) and mesoporous titania (meso-T) layers. For the planar-type device having two parallel electrodes, current–voltage (I–V) curves and pulse-driven current changes could reveal successful long-term potentiation (LTP) along with long-term depression (LTD), respectively, by the RS active mesoporous two layers for 20~100 μm length. Through mechanism characterization using chemical analysis, non-filamental memristive behavior unlike the conventional metal electroforming was identified. Additionally, high performance of the synaptic operations could be also accomplished such that a high current of 10^−6^ Amp level could occur despite a wide electrode spacing and short pulse spike biases under ambient condition with moderate humidity (RH 30~50%). Moreover, it was confirmed that rectifying characteristics were observed during the I–V measurement, which was a representative feature of dual functionality of selection diode and the analog RS device for both meso-ST and meso-T devices. The memristive and synaptic functions along with the rectification property could facilitate a chance of potential implementation of the meso-ST and meso-T devices to neuromorphic electronics platform.

## 1. Introduction

Since intensive introductions of resistive switching (RS) devices as novel memory element after Chua’s perception, many kinds of RS devices have developed and demonstrated [1,2,3,4]. The adoption of RS devices has been mainly resided on digital switching element due to first introduction of RS phenomena on ReRAM devices. Therefore, restrictions of utilization of RS devices should include a lack of a lateral electrode device with long channel structure to show digital RS switching or analog RS switching [1,2].

For the RS device, transition metal oxides (TMOs) such as TiO_2_, Ta_2_O_5_, and NiO, have been extensively adopted and successfully implemented into the RS active layers including memristors and synaptic devices. Typically, the TMO active devices have been fabricated by sputtering as well as solutional processes such as the sol-gel process [3,4].

For the RS property mechanisms of the TMO devices, filament formation from metal electrodes such as aluminum (Al) or silver (Ag) has been typically proposed for capacitor-type devices or crossbar-type devices. Usually, the electroforming process has been required for the RS behavior to show reversible formation and rupture of metallic filament, which could be observed even in lateral electrode devices upon 100~1000 nm long TiO_2_ channel [3]. In this case, physically migrated or protruded Al mass could be one or multiple electrical paths to touch the other side electrode to accomplish the RS property [4,5].

In addition, for the RS property of the planar electrode device, field-assisted activated hopping of metal ions could be a resource for the filament formation and diminution. However, its structural instability with poor retention had to block the further development for a neuromorphic device application [3].

Meanwhile, non-filamental RS behaviors have been continuously demonstrated with TMO active layers [4,5,6]. Typically, the operation mechanism of the non-filamental RS devices have been laid with formation of conduction filaments (CFs) inside of TMO materials, which were involved with evolutions of oxygen ions or oxygen gas [4]. The RS phenomena have been based on movement, accumulation, and resultant agglomeration of oxygen vacancies transited into CFs [6]. For the mechanism of CFs formation, oxygen deficient and non-stoichiometric phase has been believed to be generated in the CFs, which could be reversibly disrupted or resurrected under direct current (DC) voltage or pulsed spike voltage under electrode-stack-type or crossbar-type device structures.

Meanwhile, there have been very limited studies about lateral-electrode-formatted RS devices, which has long RS active layer between two planar electrodes. For the lateral electrode study, 0.05~0.1 μm long RS active layer between planar electrodes have been examined to elucidate and analyze the possible existence of metal filament or oxygen vacancy-based CFs through a direct tracking tool [6]. For example, with a high resolution transmission electron microscopy (HR-TEM) observation, the filament formation between two lateral electrodes could be explicitly observed, which were formed by d-orbital overlap covering electrode metal (Pt) and reduced metal ions of the TMO layer [6]. Particularly, through the real-time TEM analysis, O_2_ gas bubble formation could be identified during the oxygen vacancy build-up process inside of the TMO layer. However, the length of CFs was as short as 0.05~0.1 μm between two electrodes, which was commensurate with the thickness of stack-type or crossbar-type devices.

Separately, with the TMO-based RS devices, nanoparticles (NPs)-shaped TMO materials can be applied to fabricate the RS devices with or without a polymeric matrix [7,8]. However, the NPs-based devices were strongly dependent on the charge trapping mechanisms, which could confine the dimension of the RS phenomena, especially in length scale [7,8,9].

For the TMO RS devices, electroforming of oxygen vacancy through conduction filaments (CFs) has been widely accepted for RS phenomena mechanism of transited current level at high resistance state (HRS) up to low resistance state (LRS) [10,11,12].

Specifically, for TiO_2_-based RS devices, mesoporous silica (SiO_2_) titania (TiO_2_) (meso-ST) has been introduced recently. Mainly, the meso-ST composites have been characterized with high surface area, degree of hydrophilicity, excellent ion exchange ability, and stable framework structure, which have drawn considerable attention due to their catalytic properties [11]. However, the meso-ST could have shown memristive characteristics of long-term potentiation (LTP) associated with spike time dependent plasticity (STDP) under cap-type or crossbar-type device layouts [11]. Here, the mesoporous nature could provide higher chances of forming CFs bundles with empty spaces or rooms to facilitate formation of O_2_ gas or its diminution, which would be critical to oxygen vacancy generation or metallic cation’s reduction event.

Very recently, there have been continuous demonstrations of synaptic transistors, which should have relatively long channels (~10 μm) between the lateral source and drain electrodes. However, no sufficient discussions have been dedicated to the mechanism of memristive behaviors [12,13].

In fact, for the conduction mechanism of SiO_2_-based RS active material, ionic transport should be significantly considered as a major mechanism of current path with charge carrier of proton (H^+^) migration [12,13,14,15]. Even in these cases, the meso-ST and meso-T layers can be considered as an electrolyte-based active channel, which may facilitate fast switching of migrative ions [13,14]. For example, in mesoporous silica, ions or charge carriers could be mainly protons, which have the smallest size apart from typical charge carriers of electron or hole [13]. Therefore, ionic current would be responsible for current–voltage (I–V) curves having hysteresis behaviors.

Along with TMOs, a broad area of materials has been shown the applications in RS switching such as 2D materials including perovskite and chalcogenide with long channels [16,17]. For the perovskite, there have been many reports about dual functionality of digital and analog RS properties, which could be successfully demonstrated synaptic device operations [16]. However, there have not been many lateral device formats so far with the perovskites, instead of the vertical device format. For the perovskite case, metal ions such as Ag^+^ could migrate through the electrolyte-like perovskite layer for the RS phenomena [16]. For the case of chalcogenides such as MoS_2_, a possibility of lateral-type format for the synaptic device purpose was strongly identified [17]. The mechanism of the metal dichalcogenide device could be investigated with scanning tunneling microscope (STM), which concretely detected defects throughout longitude of the 2D layer. The origin of defects was identified as metal substitutions into sulfur vacancies, which can open up a precisely tuned defect engineering for the planar-type memristor device [17].

In this study, meso-ST and meso-T have been examined and demonstrated as memristive active layer with long channel gap (20~100 μm) between two Al lateral electrodes. Using the mesopores, more accessible O_2_ formation and diminution were available through the pores and, as a result, CFs formation through the long gap between planar electrodes could be achieved to show neuromorphic memristive characteristics. Through this study, unlike the common expectation of the catalytic application about meso-ST and meso-T matrix, a novel application to exploit synaptic device function can be drawn in integrated circuits (ICs) of brain-inspired electronics.

For comparisons of the meso-ST and meso-T with other TiO_2_ and SiO_2_-based RS materials, Table 1 is presented in terms of deposition method, unipolar or bipolar switching behavior, digital or analog characteristics, and conduction mechanisms [18,19,20]. Above all, most of the TiO_2_-based RS layers have been prepared by sputtering method so far [3,4,6,10]. Even an electron beam evaporation could be particularly utilized for the TiO_2_-based RS layer as shown in Table 1 [18]. However, both evaporation and sputtering methods require ultra-high vacuum (UHV) system to prepare the films. Meanwhile, the meso-ST and meso-T films could be prepared by a simple spin-coating method. For another comparison of the meso-ST with the SiO_2_-based RS material, SiO_2_ could be usually prepared by plasma-enhanced chemical vapor deposition (PECVD), which also requires massive film deposition equipment unlike the spin-coating method [19]. In addition, there have been reports to prepare the SiO_2_-based films by the spin coatings based on sol-gel formation [12,14,20]. However, the SiO_2_-based films were not RS-active [12], non-synaptic without an additional semiconductor [14], and solely digital functioned [20].

## 2. Materials and Methods

### 2.1. Film Formation of Mesoporous Silica–Titania and Mesoporous Titania

All chemicals were purchased from Sigma Aldrich (Burlington, MA, USA), unless specified separately. To make the meso-ST layer having amorphous or anatase TiO_2_ composition, 0.1 mol of titanium isopropoxide (Ti (OPr)_4_) was dissolved in ethanol (75 mL) while 0.1 mol citric acid (Sigma Aldrich) was dispersed in other ethanol (100 mL). The citrate acid solution was decently supplied into the Ti (OPr)_4_ solution in a dropwise manner. Then, full volume of the solution was made up to 200 mL with additional ethanol. Eventually, a final stage precursor solution of 0.5 M titanium citrate complex could be ready to be used for spin-coating [11].

Separately, surfactant of F127 (1.6 g) and 0.1 M HCl (1.0 g) was mixed into 10 mL ethanol and DI water solution (4:1). Then, tetra ethyl oxysilane (TEOS, Sigma Aldrich) (2.08 g) was supplied and subsequently provided with 0.5 M titanium citrate complex solution (4.0 mL). The final precursor mixture was turned into an orange color immediately. By 2 h stirring in ambient condition, the orange clear solution had become a glue-like solution before the complete evaporation of ethanol.

To prepare the precursor solution of meso-T layer, all the procedure was the same to the protocol of the meso-ST precursor solution except for the addition of the TEOS. Therefore, 10 mL solution of ethanol/distilled water (4:1) having F127 and 0.1 M HCl were mixed under continuous stirring to obtain a transparent solution of the meso-T precursor without TEOS. Subsequently, after drop-wise incorporation of the titania precursor solution was ready for the further procedure, the solution was then kept for vigorous stirring for 2 h. Later, 2 h of the stirring could show a fully transparent gel-like solution.

### 2.2. Fabrication of Devices

The substrates for the meso-ST and meso-T devices were 300 nm SiO_2_ covered p-type Si wafers. The 300 nm SiO_2_ was prepared by thermal dry-wet-dry oxidation at 1100 °C, which did not allow any significant leakage current.

To fabricate the meso-ST layer, after dropping of 2 mL of meso-ST precursor solution, the spin coating was carried out to have ~200 nm thickness. Then, after annealing at 300 °C for 30 min, the Al electrodes was patterned through a lift-off photolithography process as shown in Figure 1. The photoresist (DNR-L300-40, Dongjin Semichem Co., Seoul, Korea) was first spin-coated on the annealed meso-ST layer and lithography was performed to have distance of 20 μm between source and drain electrodes. The developer (MIF-300, AZ chemical, Philadelphia, PA, USA) was used in this study. Then, thermal evaporation of Al onto the photoresist patterned surface was completed to construct electrodes having an active length of 20 μm. After the Al deposition, to remove the photoresist remaining portion, acetone was used with strong sonication. The 20 μm length of active channel could not be fabricated with a simple shadow mask-aided evaporation process. For the meso-T film, the titanium citrate complex liquid was spin coated on the 300 nm SiO_2_ covered p-type Si wafer. In this case, the top electrodes of Al were deposited by thermal evaporation with source and drain patterned shadow mask, which could be accomplished in a simpler process than the lift-off process.

### 2.3. Characterization of Meso-ST and Meso-T Films

Surface morphology and film thickness were measured by a surface profiler (Veeco Co., New York, NY, USA). The elemental compositions of the meso-ST and meso-T were quantified by EDS (Energy Dispersive X-ray) spectroscopy under scanning electron microscopy (SEM, COXEM, EM-30AX, Daejeon, Korea). The film’s crystallinity was observed by X-ray diffractometer (XRD, Phillips, CuKa radiation at 1.54 nm). For the XRD analysis, meso-ST film was formed on sputtered Pt film (150 nm) on 300 nm SiO_2_ covered p-type Si wafer.

Effect of thermal treatment was measured by FT-IR (IRTracer-100, Shimadzu, Kyoto, Japan) under attenuated total reflection (ATR) mode. Raman spectroscopy (Raman 532ER, Wasatch Photonics, Logan, UT, USA) was measured in the range of 200–4000 cm^−1^. FE-SEM (field emission SEM, SU-70 Hitachi) demonstrates top-view images of mesoporous layers. Additionally, transmission electron microscopy (JEOL, JEM-ARM200F) was also used for surface mesopore images.

### 2.4. Electrical Characterization of Meso-ST and Meso-T Films

First, I–V curves were obtained using Agilent semiconductor parameter analyzer 4156C on Al parallel electrodes. The entire electrical characterizations were performed in ambient condition. The I–V curves were recorded in a sweep format by increasing the step of 0.05 V from 0 V to ±9 V. For synaptic device characterization, pulse-based I–V measurements with Agilent 4150B pulse generator were used. It was performed with the stress or spike voltage fixed at ±7 V, the stimulation delay at 0.1 s, and the stress or spike widths at 0.05 s for the meso-T device and 0.3 s for the meso-ST device, respectively. Reading currents have been obtained at read voltage (+0.1 V) for LTP and LTD synaptic operations.

## 3. Results and Discussions

Figure 1 shows device structures with long channel (22.64 μm) between the lateral electrode for the meso-ST device. As shown in Figure 1, mesopores could be identified by TEM image with the meso-ST layer. Both meso-ST and meso-T films were prepared by evaporation-induced self-assembly (EISA) method, which constructed well-ordered mesopores of ~10 nm as shown in the TEM image of Figure 1.

As shown in Figure 1, all the lateral electrodes were Al, which could be easily oxidized in contact of the meso-ST or meso-T layers. Even though there could be some oxidations on the Al surface, it was assumed that current injection was still available. In addition, since the channel length was very long in this study, the risk of oxygen or oxygen ion (O^2−^) transport could be minimized, which could be critical under the existence of aluminum oxide (AlO_x_) layer. Moreover, as illustrated in Figure 1, the meso-ST and meso-T layers were formed on an insulating layer of 300 nm thermally grown SiO_2_ surface. Therefore, there were no leakage current between electrodes and p-doped Si in these devices.

Figure 2a,b show analog non-volatile memory I–V curves as well as rectifying behavior shown for both meso-ST and meso-T devices. Even under high possibility of humidity effect on the meso-ST and meso-T devices, all the experimental results have been taken in ambient conditions. Furthermore, during each test, the relative humidity (30–90%) was recorded. The devices were stable during the measurements in ambient conditions. As shown in Figure 2a, small hysteresis could be found for I–V curves of the meso-ST device with 22.64 μm channel. The length was so long that electrodes could not trigger simple metallic migration, which eventually would form metallic filaments. Therefore, the conduction mechanism is believed to be from CFs out of oxygen vacancy and ionic or proton transport [4,6,11,14,15]. Actually, low temperature (300 °C) processed silica layer was assumed as an electrolyte film having proton-based electrodynamics [14]. Since the meso-ST layer has the low temperature annealed silica portion, the long channel could provide the memristive I–V curves based on the proton migration. The inlet graph of Figure 2a shows EDS elemental analysis, which proved that there were no Al elements in channel area after repeated memristive I–V sweeps [3]. Therefore, it is suggested that no metallic filaments were involved for the memristive I–V behaviors as identified in Figure 2a.

For Figure 2b, active channel length of the meso-T device was up to 100 μm, which could not be originated from metallic migration as well [3]. Here, due to the huge difference channel length (20 μm vs. 100 μm), the orders of current levels were varied from the meso-ST device (~10^−7^ Amp) to the meso-T device (~10^−8^ Amp) significantly. After I–V characterizations, the devices were very stable to show the memristor behaviors for one year in ambient conditions.

With an operating voltage over ± 5 V, a high current value of 10^−7^ Amp was observed for the meso-ST device’s excursive I–V curves. Here, the eight-wise excursions in Figure 2a could be another representation of oxygen-based CFs evolution mechanism as reported in previous literature [11]. Furthermore, the meso-T device in Figure 2b shows a decrease in the current level with each successive I–V cycle which implicitly plays a role in depression property in a typical neuromorphic characteristic. In addition, as shown in I–V curves, the direction of I–V hysteresis was counter-clockwise (CCW) for both meso-ST and meso-T. In addition, the I–V curves shows analog RS memory having concurrent threshold RS, which can have rectifying characteristics.

The threshold RS property can act as a selection diode, which can reduce crosstalk among RS or memristive devices under a high integration state. This rectification property could help the device to show unidirectional current flow which can be useful to mimic unidirectional synaptic property which helps the device to improve learning and unlearning phenomenon for electrical synapse [11].

Figure 3 shows pulsed spike effect of LTP and LTD for the (**a**) meso-ST device and (**b**) meso-T device, respectively. The y-axis represents currents under read voltage of +0.1 V with repeated pulse or spike operations. In the implementation of the pulse measurement, through repeated measurements of 50 times for meso-ST and 30 times for meso-T in each potentiation and depression condition, a certain tendency of synaptic characteristics was identified. In Figure 3a, the spike time width was 0.3 s with ±7 V spike voltage for the meso-ST device. However, for the meso-T device as shown in Figure 3b, long spike time width such as 0.3 s could now demonstrate discrete potentiation or depression. Instead, a very short spike time width of 0.05 s could generate noticeable synaptic potentiation and depression.

Compared with a cap-type device, the extent of LTP in Figure 3a was much smaller than the stacked meso-ST active device due to the long active length between lateral electrodes [11]. Overall, synaptic responses of the meso-ST device were more apparent than those of the meso-T device, apart from channel length difference (20 μm vs. 100 μm). The higher synaptic efficacy with the meso-ST is believed to be from ionic or proton transport through mesoporous matrix of silica [12]. At first, since the Si atoms are smaller than Ti atoms in meso-ST film, the meso-ST having silica would be more advantageous for H^+^ transport than meso-T film without silica [12].

Figure 4 shows XRD spectra for the meso-ST layer. The meso-ST revealed peaks positioned at 39.9, 46, 67.5, 82.2, and 86.1, corresponding to the planes of anatase (111), (200), (220), and rutile (311), (222) in accordance with JCPDS card no. (21–1276), additionally having hematite plane (440), respectively. Here, the anatase phase of titania is dominant probably by the fact that a small fraction of Ti atoms interacted or entered into the lattice structure and induced the lattice expansion. Here, the Si ion radius is known to be smaller than the Ti^4+^ (r = 0.079 nm). The average of crystallite size of ~9 nm is calculated for the meso-ST as-synthesized sample. It shrinks down to 8 nm after 1 h heating and again expands up to 10 nm after 2 h heating [11]. However, since the annealing temperature was still as low as 300 °C, it is believed that there is a significant portion of amorphous phase in this film.

Figure 5 shows the Raman spectra of meso-ST as-synthesized, annealed at 200 °C and 300 °C for 1 h and 2 h, respectively. The intense peaks at 785 cm^−1^ in meso-ST as-synthesized and annealed samples are ascribed to vibrational absorbance of TiO_2_ or Ti-O-Ti bonds stretching. In addition, different phases of TiO_2_ such as rutile and anatase showed strong absorption bands in range of 650~850 cm^−1^ [11]. However, in Figure 5, there are no corresponding peaks for rutile or anatase state, which could suggest an amorphous state of the m-ST. Sharp bands at 1080~1083 cm^−1^ in separate samples are attributed to asymmetric Si-O-Si vibrational signals, which are very weak for the as-synthesized sample. Meanwhile, edge peak at 1197 cm^−1^ on three meso-ST samples correspond to asymmetric stretching vibrations of Si-O-Si [11]. Additionally, there are no peaks detected for bulk Si or crystalline Si in Figure 5 [21].

In the meso-ST layer, broad peaks around 1203 cm^−1^ and 1371 cm^−1^ are attributed to the O-C=O symmetric and asymmetric stretch modes. They represent organic functional groups or volatile compounds of surfactant F127. The bands could be diminished with further annealing as shown in Figure 5. A noticeable peak at 1741 cm^−1^ in meso-ST as-synthesized sample is ascribed to C=O bond’s stretching or vibration, which decreased after 2 h annealing. The bands at 2876 cm^−1^, 2946 cm^−1^, and 2279 cm^−1^ of the meso-ST can be attributed to the C-H stretching bonds, respectively, which are constituents of organic surfactant.

The bands around 1655 cm^−1^, which shifted to 1631 cm^−1^ in three meso-ST layers, are assigned to bending or vibration mode of H-O-H. The peak for the H-O-H can be from a protonated silanol group, which can greatly influence the proton transport under an electrical field [12,14,15]. The peculiar band at 3283 cm^−1^ (O-H stretching) in the meso-ST became a gradually broader form as-synthesized to annealed at 3013 cm^−1^, which was ascribed to the presence of Si-OH and Ti-OH bonds. Particularly, the silanol group (Si-OH) is believed to be critical to ionic current by surface-mediated transport depending upon its degree of protonation (-O^+^H_2_) within mesopores [12]. The annealed meso-ST is known to be evolved into solid semi-spheres, which showed the presence of mesopores (~10 nm diameter) among the particles. The generated mesopores would allow O_2_ gas to penetrate within the meso-ST’s interior by diffusion, which can influence the conduction mechanism of the meso-ST layer.

## 4. Conclusions

In this study, we have introduced synaptic device functions with meso-ST and meso-T active layers with long channel lateral electrodes. Asymmetric pinched hysteresis approves the self-rectification with different threshold voltages, which was significant to RS-based memory devices as it evaded the cross talk. Both devices showed potentiation and depression synaptic properties. In the near future, the meso-ST and meso-T devices are expected to be successfully integrated in planar structured neuromorphic electronics.

## Figures and Tables

**Figure 1 nanomaterials-13-01734-f001:**
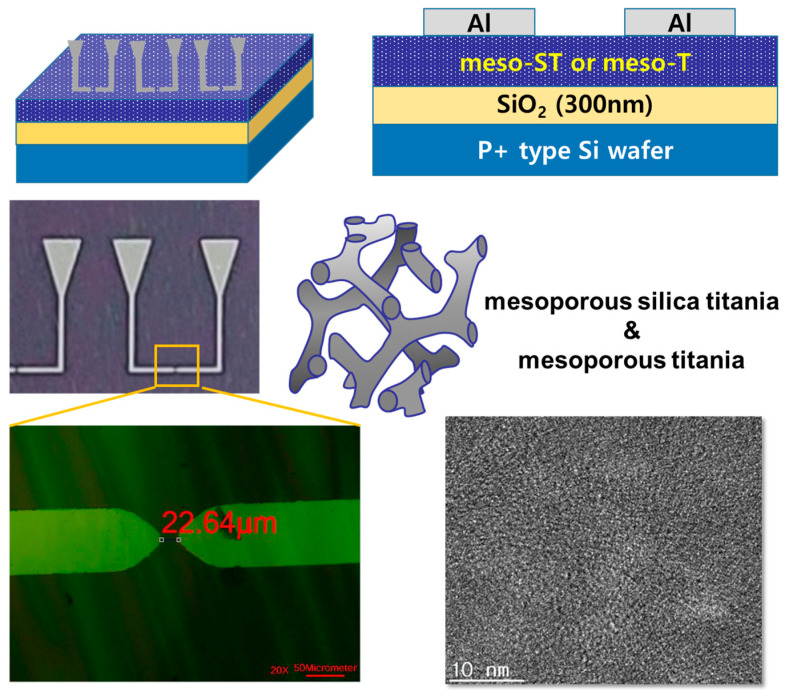
Device structure with long channel (22.64 μm) between lateral electrode for meso-ST device and TEM image of the meso-ST surface.

**Figure 2 nanomaterials-13-01734-f002:**
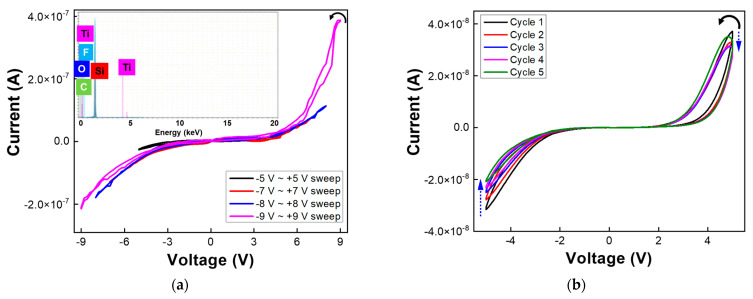
IV characteristics for (**a**) meso-ST at ±5 V~ ±9 V voltage (**b**) meso-T for five repetitive cycles at ±5 V.

**Figure 3 nanomaterials-13-01734-f003:**
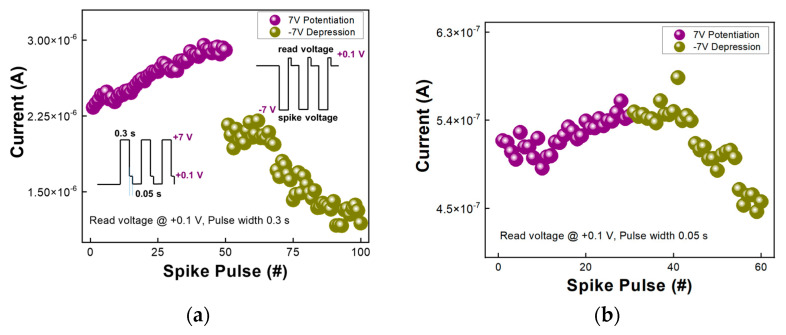
LTP behavior subsequent with LTD for (**a**) meso-ST at peak voltage ±7 V in pulse width 0.3 s and (**b**) meso-T at read voltage +0.1 V in pulse width of 0.05 s.

**Figure 4 nanomaterials-13-01734-f004:**
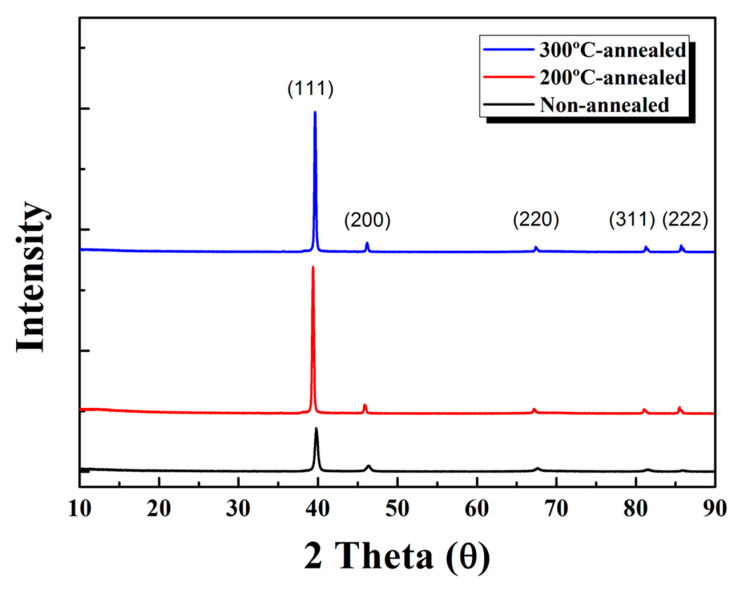
XRD spectra of the meso-ST surface depending on annealing temperature.

**Figure 5 nanomaterials-13-01734-f005:**
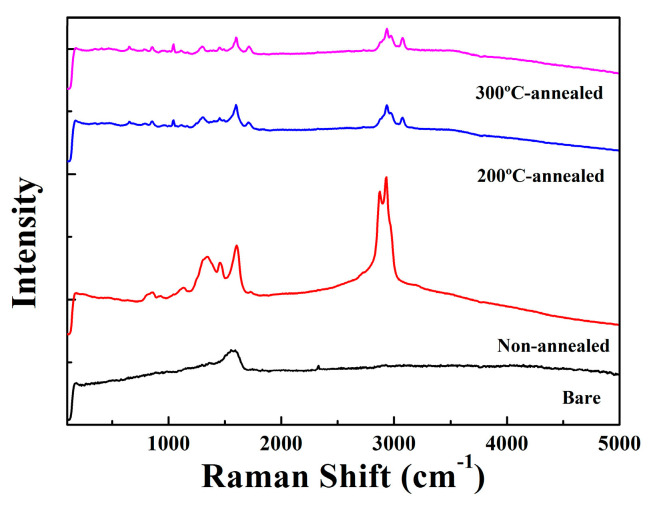
Raman spectra of the meso-ST surface depending on annealing temperature.

**Table 1 nanomaterials-13-01734-t001:** Comparison of RS property characteristics of TiO_2_ and SiO_2_ based device.

Material	Deposition Method	Unipolar/Bipolar Switching	Digital/Analog Application	Conduction Mechanism	Reference
TiO_2_	Electron beam evaporation	Bipolar	Digital	Oxygen filament	[18]
SiO_2_	Plasma-enhanced chemical vapor deposition	Unipolar	Analog	Oxygen filament	[19]
TiO_2_/ZrO_2_	Spin coating	Bipolar	Digital	Oxygen filament	[20]
meso-ST or meso-T	Spin coating	Bipolar	Analog	Proton transfer	This study

## Data Availability

The data presented in this study are available on request from the corresponding author. The data are not publicly available due to private intellectual property.

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
