# Peer review of "Characteristics of Synaptic Function of Mesoporous Silica–Titania and Mesoporous Titania Lateral Electrode Devices"

_nanomaterials, 2023, doi:10.3390/nano13111734_

Round 1
Reviewer 1 Report
Search for materials for memristor applications is a truly hot topic in many laboratories.
1. There is a broad selection of memristor materials e.g. based on transition metal dichalcogenide e.g. D. Akinwande Nature Nanotechnology v.16, pages58–62, 2020, or tri-chalcogenides e.g. Guihua Zhao et al 2020 Appl. Phys. Express 13 105001 , which were not discussed at all. I couldn’t find any comparison showing advantage of discussed in the manuscript materials over other materials.
2. Device fabrication section is unclear
A. I don’t understand :
The substrate for devices was 300 nm SiO2 covered p-type Si wafers. To fabricated meso-ST layer, after dropping of 2 mL of meso-ST precursor solution the spin coating is 132 carried out to have ~200 nm thickness. Then, after annealing at 300 °C for 30 min., the Al electrode is formed through a lift-off lithography process.
How it is possible to form electrodes by annealing without preceding metal deposition?
B. Which lithography technique was used to make Al contacts on first sample: photolithography or ebeam lithography or shadow deposition (as in case of a the second sample). I guess aluminum was deposited somehow. Please writte details of the fabrication process.
C. Regarding the text: For meso-T film, the titanium citrate complex liquid was spin coated. The top 135 electrode Al was deposited by metal evaporation method with source and drain patterned shadow mask.
What technique was used for Al deposition?
3. Application of Al contacts on these metals is a risky strategy. Taking into account that oxygen can move easily, aluminum can oxidize forming additional insulating layer at the interface Al/sample. It is very difficult to exclude the effect during sample fabrication process. Was this phenomenon taken into consideration at all?
4. Authors present results of charge carrier transport of meso-ST and mesoporous titania deposited on p-doped Si/SiO2 substrate. Measured currents were of order of 10-7 A – 10-8 A. Authors didn’t took into consideration that leakage current between electrodes and p-doped Si substrate is of order of 10-8 A too, which it is well known fact. So in my opinion the leakage current may significantly contribute to IV results presented in the manuscript.
5. EDS results in an inset of fig.2a is unreadable.
6. Does X-axis in fig.3 presents number of pulse?
7. It is really striking why Raman spectra shown in fig.5 did not reveal Si peak, which should be observed at about 525cm-1?

Author Response
Reply and Response (Nanomaterials 2341865)
*All corrected paragraphs and Figures in revised manuscript were tracted.
Corrected parts were highlighted in blue color, at the same time, “track changes” function was in mode during revision.
Reviewer #1 :
Search for materials for memristor applications is a truly hot topic in many laboratories.
- There is a broad selection of memristor materials e.g. based on transition metal dichalcogenide e.g. D. Akinwande Nature Nanotechnology v.16, pages 58–62, 2020, or tri-chalcogenides e.g. Guihua Zhao et al 2020 Appl. Phys. Express 13 105001, which were not discussed at all. I couldn’t find any comparison showing advantage of device discussed in the manuscript materials over other materials.
Response: Authors would like to thank reviewer for a kind comment. After appraising the comments, we have discussed about the advantages of our synthesized materials over other materials we have included the reference literatures in revised article. Also, we have referred the articles suggested by the reviewer.
In addition, Table 1 was added to show advantage of device in this study.
|
Material |
Deposition method |
Unipolar/bipolar switching curve |
Digital/ analog application |
Conduction mechanism |
Reference |
|
TiO2 |
Electron beam evaporation |
Bipolar switching |
Digital |
Oxygen filament |
18 |
|
SiO2 |
plasma-enhanced chemical vapor deposition |
Unipolar switching |
Analog |
Oxygen filament |
19 |
|
TiO2/ZrO2 |
Spin coating |
Bipolar |
Digital |
Oxygen filament |
20 |
|
meso-ST |
Spin coating |
bipolar |
Analog |
Proton transfer |
This study |
Manuscript was revised in lines of 119-132:
“For comparisons of the meso-ST and meso-T with other TiO2 and SiO2 based RS materials, Table 1 is presented in terms of deposition method, unipolar or bipolar switching behavior, digital or analog characteristics, and conduction mechanisms [18-20]. Above all, most of the TiO2 based RS layers have been prepared by sputtering method so far [3,4,6,10]. Even an electron beam evaporation could be particularly utilized for the TiO2 based RS layer as shown in Table 1 [18]. However, both evaporation and sputtering methods require ultra-high vacuum (UHV) system to prepare the films. Meanwhile, the meso-ST and meso-T films could be prepared by a simple spin-coating method. For another comparison of the meso-ST with the SiO2 based RS material, SiO2 could be usually prepared by plasma-enhanced chemical vapor deposition (PECVD), which also requires massive film deposition equipment unlike the spin-coating method [19]. In addition, there have been reports to prepare the SiO2 based films by the spin coatings based on sol-gel formation [12,14,20]. However, the SiO2 based films were not RS-active [12], non-synaptic without additional semiconductor [14], and solely digital-functioned [20].”
- Device fabrication section is unclear
- I don’t understand :
The substrate for devices was 300 nm SiO2 covered p-type Si wafers. To fabricated meso-ST layer, after dropping of 2 mL of meso-ST precursor solution the spin coating is carried out to have ~200 nm thickness. Then, after annealing at 300 °C for 30 min., the Al electrode is formed through a lift-off lithography process.
How it is possible to form electrodes by annealing without preceding metal deposition?
Response: Authors would like to thank reviewer for a kind comment. For the fabrication of the device the meso-ST was spun coated on the SiO2 wafer and procedure is then followed with annealing of the active layer. Then aluminum (Al) was deposited with thermal evaporation method. Photoresist itself along with desired photomask was patterned using lithography, subsequently, Al evaporation process was done. It provides under size of 100 micrometer objects, which cannot be prepared by shadow mask. As our research was dealing with non-filament conduction, it was favorable to have smaller channel length between the lateral electrodes. Therefore, lift-off lithography method was designated for the meso-ST. The necessary changes have been made in revised article. In addition, Figure 1 was corrected for more explanations.
Manuscript was revised in lines of 119-132:
“The substrates for the meso-ST and meso-T devices were 300 nm SiO2 covered p-type Si wafers. To fabricate the meso-ST layer, after dropping of 2 mL of meso-ST precursor solution, the spin coating was carried out to have ~200 nm thickness. Then, after annealing at 300 °C for 30 min, the Al electrodes was patterned through a lift-off photolithography process as shown in Figure 1. The photoresist (DNR-L300-40, Dongjin Semichem Co., Korea) was first spin-coated on the annealed meso-ST layer and lithography was performed to have distance of 20 mm between source and drain electrodes. The developer (MIF-300, AZ chemical, USA) was used in this study. And then, thermal evaporation of Al onto the photoresist patterned surface to construct electrodes having active length of 20 mm. After the Al deposition, to remove the photoresist remained portion, acetone was used with strong sonication. The 20 mm length of active channel could not be fabricated with a simple shadow mask-aided evaporation process. For the meso-T film, the titanium citrate complex liquid was spin coated on the 300 nm SiO2 covered p-type Si wafer. In this case, the top electrodes of Al were deposited by thermal evaporation with source and drain patterned shadow mask, which could be accomplished in simpler process than the lift-off process.”
- Which lithography technique was used to make Al contacts on first sample: photolithography or e beam lithography or shadow deposition (as in case of a the second sample). I guess aluminum was deposited somehow. Please write details of the fabrication process.
Response: Authors would like to thank reviewer for a kind comment. We have used the photolithography technique for the patterning of the top electrode (Al). The lift off photolithography method was used where photoresist was removed living behind the metal pattern. In view of the comment, we have included the detailed fabrication process in the article.
Manuscript was revised in lines of 119-132:
“The substrates for the meso-ST and meso-T devices were 300 nm SiO2 covered p-type Si wafers. To fabricate the meso-ST layer, after dropping of 2 mL of meso-ST precursor solution, the spin coating was carried out to have ~200 nm thickness. Then, after annealing at 300 °C for 30 min, the Al electrodes was patterned through a lift-off photolithography process as shown in Figure 1. The photoresist (DNR-L300-40, Dongjin Semichem Co., Korea) was first spin-coated on the annealed meso-ST layer and lithography was performed to have distance of 20 mm between source and drain electrodes. The developer (MIF-300, AZ chemical, USA) was used in this study. And then, thermal evaporation of Al onto the photoresist patterned surface to construct electrodes having active length of 20 mm. After the Al deposition, to remove the photoresist remained portion, acetone was used with strong sonication. The 20 mm length of active channel could not be fabricated with a simple shadow mask-aided evaporation process. For the meso-T film, the titanium citrate complex liquid was spin coated on the 300 nm SiO2 covered p-type Si wafer. In this case, the top electrodes of Al were deposited by thermal evaporation with source and drain patterned shadow mask, which could be accomplished in simpler process than the lift-off process.”
- Regarding the text: For meso-T film, the titanium citrate complex liquid was spin coated. The top 135 electrode Al was deposited by metal evaporation method with source and drain patterned shadow mask.
What technique was used for Al deposition?
Response: Authors would like to thank reviewer for a kind comment. In the case of meso-T films, the titanium citrate solution was spun coated on silicon wafer, to remove the impurities, the synthesis was followed with the annealing process at 300°C for one hour. On the surface of meso-T top electrode Al was deposited with thermal evaporation method using shadow mask. Desired changes have been made in the revised manuscript.
- Application of Al contacts on these metals is a risky strategy. Taking into account that oxygen can move easily, aluminum can oxidize forming additional insulating layer at the interface Al/sample. It is very difficult to exclude the effect during sample fabrication process. Was this phenomenon taken into consideration at all?
Response: Thank you for that reviewer has raised very important point. However, we believe in our case, Al as top electrode would help proton transfer through the channel. Even though there could be somewhat oxidations on Al surface, it was assumed that current injection was still available. In addition, since the channel length was very long in this study, the risk of oxygen or oxygen ion (O2-) transport could be minimized, which could be critical under existence of aluminum oxide (AlOx) layer. However, we agree that Al easily can get oxides with prolongated time and, eventually, it might form insulating layer at the interface. Nonetheless, our study is more inclined towards the protonic transfer depending on the channel length.
Manuscript was revised in lines of 204-209:
“As shown in Fig. 1, all the lateral electrodes were Al, which could be easily oxidized in contact of the meso-ST or meso-T layers. Even though there could be somewhat oxidations on Al surface, it was assumed that current injection was still available. In addition, since the channel length was very long in this study, the risk of oxygen or oxy-gen ion (O2-) transport could be minimized, which could be critical under existence of aluminum oxide (AlOx) layer.”
- Authors present results of charge carrier transport of meso-ST and mesoporous titania deposited on p-doped Si/SiO2 substrate. Measured currents were of order of 10-7 A – 10-8 A. Authors didn’t take into consideration that leakage current between electrodes and p-doped Si substrate is of order of 10-8 A too, which it is well known fact. So, in my opinion the leakage current may significantly contribute to IV results presented in the manuscript.
Response: Authors would like to thank reviewer for a critical comment. Authors provide a new diagram of device structures in Figure 1, which shows that there were no direct contacts of the meso-ST and meso-T layers to p-doped Si wafer. That is, the meso-ST and meso-T layers were formed on insulating layer of 300 nm SiO2 surface. Therefore, there were no leakage current between electrodes and p-doped Si in these devices.
Manuscript was revised in lines of 209-211:
“Moreover, as illustrated in Fig. 1, the meso-ST and meso-T layers were formed on insulating layer of 300 nm thermally grown SiO2 surface. Therefore, there were no leakage current between electrodes and p-doped Si in these devices”
- EDS results in an inset of fig.2a is unreadable.
Response: Authors would like to thank reviewer for a kind comment. The EDS figure was enlarged and captions for elements like Ti, Si, O, F, and C were enlarged in revised version.
- Does X-axis in fig.3 presents number of pulse?
Response: Thank you for reviewer’s important comment. In Figure 3, x-axis represents the number of pulse (or spike) voltage and read voltage biases given to the device. Figure 3(a) and 3(b) designates the 50 and 30 consecutive pulses and read voltages sequences assigned to the device, which in turn represents the LTP (long term potentiation) and LDP (long term depression) neuromorphic characteristics. For more clarity about pulse measurement procedure, Figure 3(a) was redrawn with pulse measurement details.
- It is really striking why Raman spectra shown in fig.5 did not reveal Si peak, which should be observed at about 525 cm-1?
Response: Authors would like to thank reviewer for a kind comment. First, the Raman spectroscopy, which authors used in this study, might have a low resolution specifically for short wavenumber. Second, since the Si atoms are mostly bonded to oxygen, not to Si atoms by themselves, authors believe that there is no Si peak for Raman spectroscopy, here.
Reviewer #2 :
The article corresponds to the subject of the journal. It is devoted to the study of the electrical properties of thin polycrystalline films (TiO2 and TiO2/SiO2). The paper discusses the possibility of observing memristive properties in the systems under study.
Response: Authors would like to thank reviewer for kind comments.
The presented version of the article raises many questions regarding the method of formation of the device under study and the obtained measurement results.
Response: Authors would like to thank reviewer for kind comments. As specified by the reviewers, the synthesis method and the measurement study have discussed in detail. We have included the schematic diagram in reviewed article. Thank you so much pointing this out.
- it should be clarified how the mesoporous structure of the films was determined, what are the characteristic sizes of mesopores, and what is the homogeneity of the obtained material.
Response: Authors would like to thank reviewer for kind comments. To find out porous structure of the fabricated thin film, we have taken TEM image and films were evenly distributed to all over the surface of substrate with well-defined pore structure. The pore size should be the rage of ~10 nano size. So, there was revised description.
Manuscript was revised in lines of 201-203:
“Both meso-ST and meso-T films were prepared by evaporation induced self-assembly (EISA) method, which constructed well-ordered mesopores of ~10 nm as shown in the TEM image of Fig. 1.”
- you need to draw a diagram of the device: the arrangement of layers, applied contacts and the characteristic dimensions of its components. It is advisable to attach a photo of the product used in the experiments.
Response: Authors would like to thank reviewer for kind comments. For the diagram of device, Figure 1 was redrawn to include the device’s structure in revised version. And, the device photo already exists in Fig. 1. The experimental device shows basic MIM (metal-insulator-metal) structure for the memristor. Where the top electrodes were aluminum, which were placed in planar surface. For our device active layers are (a) meso-ST and (b) meso-T had spun coated on 300 nm SiO2 covered silicon wafer.
- what does the length of 22.64 microns mean, how was it determined with such accuracy, and how is it related to the structural elements of the film? What do the pictures (Figure 1) mean and what can you tell about the sample based on the TEM image shown (Figure 1)
Response: Authors would like to thank reviewer for careful comments. The optical microscope image shows the channel length (width) present between lateral electrodes; the width is 22.64 micron. The distance between parallel electrodes was originally designed for 20 mm through the lift-off method. However, the real channel lengths after fabrication processes had variations. Our article discusses non-filament conduction mechanisms unlike the conventional electroforming mechanism. Hence, the distance between these lateral electrode plays an important role for the RS behavior for the proton transfer. So, except for the specific cases, the 22.64 was not highlighted nor emphasized in the revised version.
- the degree of crystallinity of the deposited film samples depends on the calcination temperature (Figure 4). Is it possible to state that the samples calcined at 300 C do not contain an amorphous phase?
Response: Authors would like to thank reviewer for critical comments. It is true that temperature can affect the crystallinity of the material. It is well known fact to get fully crystalline silicon there is requirement of high temperature. The device has showed the polycrystalline orientation. Furthermore, the size of the crystals could affect by the temperate variation. Yet, we could not neglect fact that at the low temperature there might a lowest possibility amorphous phase. Therefore, the manuscript was corrected as follows:
In lines of 288-289:
“However, since the annealing temperature was still low as 300 oC, it is believed that there is significant portion of amorphous phase in this film.”
- it is necessary to clarify whether the studied films are in contact with air. The adsorption of water, CO2, and other gases can significantly affect their electrophysical properties.
Response: Authors would like to thank reviewer for kind comments. Even under high possibility of humidity effect on the meso-ST and meso-T devices, all the experimental results have been taken in ambient conditions. Furthermore, during each test, the relative humidity (30-90%) were recorded. The devices were stable during the measurements in ambient conditions.
In lines of 191-192, 213-216:
“The entire electrical characterizations were performed in ambient condition.”
“Even under high possibility of humidity effect on the meso-ST and meso-T devices, all the experimental results have been taken in ambient conditions. Furthermore, during each test, the relative humidity (30-90%) were recorded. The devices were stable dur-ing the measurements in ambient conditions.”
6 it is not very clear what applied voltage (probably 0.1 V?) corresponds to the currents indicated in Figure 3. What is the reason for the strong change in currents after a cyclic pulse action on the sample.
Response: Authors would like to thank reviewer for kind comments. For the LTP and LTD (long term potentiation/depression) spike/pulse is provided at peak voltage simultaneously the generated pulse was read at read voltage 0.1V. Here, the peak voltage was +7V and -7V for LTP and LTD, respectively. The strong change in the current refers to the potentiation could be considered as a learning process of the device. When the spike is imported with width time along with the voltage that generates the learning ability of the device, and each spike leads the current level increase.
So, authors corrected Figure 3(a) to illustrate the scheme of potentiation and depression measurement condition.
- it would make sense to carry out X-ray phase analysis of the samples after electrophysical experiments.
Response: Authors would like to thank reviewer for kind comments. In the case of meso-ST, the consecutive I-V cycles shows stability of the device with different electric field. On the other hand, our second device have showed stability for consecutive cycles. In addition, there is no X-ray phase analysis tool nearby to scan the long channel length, unfortunately.
- Based on the data presented in the article, it is difficult to assess the stability of the synthesized films and the possibility of reproducing the effects described in it.
Response: Authors would like to thank reviewer for kind comments. The devices were very stable to show the memristor behaviors for one year in ambient conditions.
In lines of 231-233:
“After I-V characterizations, the devices were very stable to show the memristor behaviors for one year in ambient conditions.”
The article needs significant revision.
Response: Authors would like to thank reviewer for many kind comments. And, based on the comments, many ambiguities in the manuscript could be eliminated and significantly revised to extend the length of script to ensure more accurate descriptions.

Reviewer 2 Report
The article corresponds to the subject of the journal. It is devoted to the study of the electrical properties of thin polycrystalline films (TiO2 and TiO2/SiO2). The paper discusses the possibility of observing memristive properties in the systems under study.
The presented version of the article raises many questions regarding the method of formation of the device under study and the obtained measurement results.
- it should be clarified how the mesoporous structure of the films was determined, what are the characteristic sizes of mesopores, and what is the homogeneity of the obtained material.
- you need to draw a diagram of the device: the arrangement of layers, applied contacts and the characteristic dimensions of its components. It is advisable to attach a photo of the product used in the experiments.
- what does the length of 22.64 microns mean, how was it determined with such accuracy, and how is it related to the structural elements of the film? What do the pictures (Figure 1) mean and what can you tell about the sample based on the TEM image shown (Figure 1)
- the degree of crystallinity of the deposited film samples depends on the calcination temperature (Figure 4). Is it possible to state that the samples calcined at 300 C do not contain an amorphous phase?
- it is necessary to clarify whether the studied films are in contact with air. The adsorption of water, CO2, and other gases can significantly affect their electrophysical properties.
- it is not very clear what applied voltage (probably 0.1 V?) corresponds to the currents indicated in Figure 3. What is the reason for the strong change in currents after a cyclic pulse action on the sample.
- it would make sense to carry out X-ray phase analysis of the samples after electrophysical experiments.
Based on the data presented in the article, it is difficult to assess the stability of the synthesized films and the possibility of reproducing the effects described in it.
The article needs significant revision.
Author Response

(The authors gave the same response as above.)

Round 2
Reviewer 1 Report
Authors would like to thank reviewer for a critical comment. Authors provide a new diagram of device structures in Figure 1, which shows that there were no direct contacts of the meso-ST and meso-T layers to p-doped Si wafer. That is, the meso-ST and meso-T layers were formed on insulating layer of 300 nm SiO2 surface. Therefore, there were no leakage current between electrodes and p-doped Si in these devices.
I do not agree with the explanation. Of course electrodes were not contacted with the p-doped Si but with 300nm thick SiO2. The leakage current leaks thorough the SiO2 layer into the Si substrate see for example:
1. Characterization of thin Al2O3/SiO2 dielectric stack for CMOS transistors, Yiyi Yan et.al. Microelectronic Engineering v. 254, 111708, 2022, doi:10.1016/j.mee.2022.111708
2. Study on the low leakage current of an MIS structure fabricated by ICP-CVDS-Y Tsai et.al. Journal of Physics: Conference Series, v. 100, ELECTRONIC MATERIALS AND PROCESSING 2008 J. Phys.: Conf. Ser. 100 042030, 2008
Authors didn’t write dimensions of deposited aluminium contacts, which determine the leakage current.
The electric current values reported in the paper include both leakage current and current transmitted through the sample. Therefore, without determination of the leakage current, the results are unreliable.
Authors would like to thank reviewer for a kind comment. First, the Raman spectroscopy, which authors used in this study, might have a low resolution specifically for short wavenumber. Second, since the Si atoms are mostly bonded to oxygen, not to Si atoms by themselves, authors believe that there is no Si peak for Raman spectroscopy, here.
I do not agree with this answer. Silicon is a standard sample used for calibration of Raman spectrometers. Even Si substrates with 300nm thick SiO2 can be used for such calibration.
Lack of a Si-related peak in the Raman spectrum presented in the manuscript is a signature of badly calibrated spectrometer. In fig.1 one can clearly see Si substrate. A laser beam diameter for is large enough to collect spectra both from your sample and from the SiO2.
Author Response
Reply and Response (Nanomaterials 2341865)
*All corrected paragraphs and Figures in revised manuscript were tracted.
Corrected parts were highlighted in maroon color, at the same time, “track changes” function was in mode during revision.
Reviewer #1 :
Authors would like to thank reviewer for a critical comment. Authors provide a new diagram of device structures in Figure 1, which shows that there were no direct contacts of the meso-ST and meso-T layers to p-doped Si wafer. That is, the meso-ST and meso-T layers were formed on insulating layer of 300 nm SiO2 surface. Therefore, there were no leakage current between electrodes and p-doped Si in these devices.
I do not agree with the explanation. Of course electrodes were not contacted with the p-doped Si but with 300nm thick SiO2. The leakage current leaks thorough the SiO2 layer into the Si substrate see for example:
Response: Authors would like to thank reviewer for double-check comment. First, the lateral electrodes in author’s study, there is no electric field across the 300 nm SiO2 layer during measurement. However, reviewer is worried about very thin ( < 4 nm) or not-thermally grown oxide or leaky oxide deposited by CVD at low temperature cases and both cases were under vertical electric fields across the silicon dioxides. Second, for thermally grown oxide even with 300 nm thickness, there is a very low chance to have significant leakage, unless severe chemical or physical damages existed. In this study, the 300 nm SiO2 layer was firmly prepared in dry oxidation (with O2 gas at 1100 oC) + wet oxidation (with steam at 1100 oC) + dry oxidation (with O2 gas at 1100 oC), which means that the oxide had low leakage current. Again, the two reference papers were under special situations like (1) in 4 nm or under thickness layer and with Al2O3 layer, (2) ICP-CVD layer, which can have leakage current.
Separately, for a rainy case, we measured current (Ip-type Si) under contact of the p-typed Si side, the current level was not in 10-9 Amp level as reviewer worried about. It was 10-15~10-16 Amp levels under 5 V between two lateral electrodes, which was under lateral electric field not passing through p-type Si wafer.
- Characterization of thin Al2O3/SiO2 dielectric stack for CMOS transistors, Yiyi Yan et.al. Microelectronic Engineering v. 254, 111708, 2022, doi:10.1016/j.mee.2022.111708
Response: The paper which reviewer provided has vertical or stacked device with SiO2 layer. However, in author’s study, the device has lateral electrodes, which means that there is no electrical field through the 300 nm SiO2 in author’s study.
- Study on the low leakage current of an MIS structure fabricated by ICP-CVDS-Y Tsai et.al. Journal of Physics: Conference Series, v. 100, ELECTRONIC MATERIALS AND PROCESSING 2008 J. Phys.: Conf. Ser. 100 042030, 2008
Response: The paper which reviewer provided had vertical or stacked device with SiO2 layer. However, in author’s study, the device has lateral electrodes, which means that there is no electrical field through the 300 nm SiO2 in author’s study.
In addition, the quality of thermally grown SiO2 cannot be compared with CVD grown SiO2, even at low temperature as 90 oC.
Authors didn’t write dimensions of deposited aluminium contacts, which determine the leakage current.
Response: Again, there is no electric field across the SiO2 layer.
The electric current values reported in the paper include both leakage current and current transmitted through the sample. Therefore, without determination of the leakage current, the results are unreliable.
Response: The very thin layer (under 4 nm) of SiO2 layer as reviewer’s example paper 1 can have tunneling current. Author’s group have had many experiences about the tunneling oxide (~ 10 nm thickness) as published in Appl. Phys. Lett. 97 (2010) 153302. However, in author’s study of this revised paper, we used 300 nm, which is 30 times thicker.
Manuscript was revised in lines of 160-161:
“The 300 nm SiO2 was prepared by thermal dry-wet-dry oxidation at 1100 oC, which did not allow any significant leakage current.”
Authors would like to thank reviewer for a kind comment. First, the Raman spectroscopy, which authors used in this study, might have a low resolution specifically for short wavenumber. Second, since the Si atoms are mostly bonded to oxygen, not to Si atoms by themselves, authors believe that there is no Si peak for Raman spectroscopy, here.
I do not agree with this answer. Silicon is a standard sample used for calibration of Raman spectrometers. Even Si substrates with 300nm thick SiO2 can be used for such calibration.
Lack of a Si-related peak in the Raman spectrum presented in the manuscript is a signature of badly calibrated spectrometer. In fig.1 one can clearly see Si substrate. A laser beam diameter for is large enough to collect spectra both from your sample and from the SiO2.
Response: First, in author’s paper, silica was formed in the meso-ST film not bulk Si or crystalline Si, which can have strong Si-Si bond. In addition, authors do not understand what is meaning of standard sample for Raman with Si. There have been many reports using Raman spectroscopy without using Si as a reference, especially for silica or SiO2 based materials. In fact, authors found many Raman spectra data in following papers, which did not have or represent or provide 525 cm-1 peak with using silicon or silica containing materials. They are as shown below:
- Mater. Chem. A (2016), 4, 4570. Graphene and silica material case. They did not comment any 525 cm-1 peak.
Phys. Chem. Chem. Phys. (2014) (DOI: 10.1039/c4cp03582h) rhodamine 6G deposited Si wafer. Even though they used bulk Si of Si wafer, they never comment 525 cm-1 around peak.
Mater. Lett. (2016), 170, 179-182. In the case of silica, not in the case of bulk-Si, there was no Raman peak at 525 cm-1 peak.
Nano. Res. Lett. (2017), 12, 292. There was no 525 cm-1 peak for SiO2 and carbon powder.
Only several paper showing bulk-Si or crystalline Si moiety (not amorphous silica) could show around 525 cm-1 peak.
- Mater. Chem. A. (2016) 4, 6098. For strong reduced by Mg porous Si, accompanied by greatly reduced size, there are crystallinity and Raman peak around 500 cm-1, even not at 525 cm-1.
One paper clearly assigned the 525 cm-1 peak of Raman for crystalline-Si.
2013 Conference on Lasers & Electro-Optics Europe & International Quantum Electronics Conference CLEO EUROPE/IQEC, title: “Photon-counting Raman spectroscopy of silicon nanowires”
Manuscript was revised in lines of 301-302:
“Additionally, there are no peaks detected for bulk Si or crystalline Si in Fig. 5 [21].”
Reviewer #2 :
The authors have significantly improved the article. In the presented form, the article can be published.
Response: Authors would like to thank reviewer for kind evaluation.

Reviewer 2 Report
The authors have significantly improved the article. In the presented form, the article can be published.
Author Response
Reply and Response (Nanomaterials 2341865)
Reviewer #2 :
The authors have significantly improved the article. In the presented form, the article can be published.
Response: Authors would like to thank reviewer for kind evaluation.

Round 3
Reviewer 1 Report
It was 10-15~10-16 Amp levels under 5 V between two lateral electrodes, which was under lateral electric field not passing through p-type Si wafer.
One can determine the leakage current just measureung the current as a function of gate voltage Ug and extrapolate the value of the leakage current for Ug=0V. This has not been done. Instead Authors wrote that they measured the leakage current of order of 10^-15 - 10^-16. According to catalogue data the instrument mentioned in the paper (Agilent semiconductor parameter analyzer 4156C) is not capable of measuring currents of 10^-15 - 10^-16 A.
The Si peak at about 521cm-1 has very large intensity. The peak might not be present on collected Raman spectra under the condition, that intensities of Raman spectra of the meso-ST are much larger than the intensity of the Si peak. Lack of Raman intensity in Raman spectra presented in the manuscript disables to judge the intensities of spectra presented in fig.5.
Still I am not convinced of the correctness of the execution of the Raman experiment.
Authors referred to following papers claiming, that the Si peak in Raman spectra doesn’t have to be present. But:
1. In Nano. Res. Lett. 12 292 2017 they do refer to the Si peak (fig.2b)
2. In 10.1039/c4cp03582h authors didn’t show at all the Raman shift range, where the Si peak should appear.
3. I measured myself the rhodamine in top of SiO2 and I could see the Si peak.
Still I am not convinced of the correctness of the execution of the Raman experiment.
Quality of the inset in the fig2a is unacceptable.
Author Response
1. It was 10-15~10-16 Amp levels under 5 V between two lateral electrodes, which was under lateral electric field not passing through p-type Si wafer. One can determine the leakage current just measureung the current as a function of gate voltage Ug and extrapolate the value of the leakage current for Ug=0V. This has not been done.
=> It can be done with Agilent semiconductor parameter 4156C. We use a measurement program of sagatorius. Anyway, there is no leakage current to bottom Si substrate. Please refer to the diagram which authors suggested.
Instead Authors wrote that they measured the leakage current of order of 10^-15 - 10^-16. According to catalogue data the instrument mentioned in the paper (Agilent semiconductor parameter analyzer 4156C) is not capable of measuring currents of 10^-15 - 10^-16 A,
=> It can be done with Agilent semiconductor parameter 4156C apparently with 1 femto Amp.
Authors don't have any idea why reviewer's manual has different descriptions.
And, at any rates, we are having femto measurements.
https://www.keysight.com/us/en/product/4156C/precision-semiconductor-parameter-analyzer.html
https://www.valuetronics.com/product/4156c-agilent-semiconductor-parameter-analyzer-used
and more
Authors don't believe that it is a really important reviewing issue.
2. The Si peak at about 521cm-1 has very large intensity. The peak might not be present on collected Raman spectra under the condition, that intensities of Raman spectra of the meso ST are much larger than the intensity of the Si peak. Lack of Raman intensity in Raman spectra presented in the manuscript disables to judge the intensities of spectra presented in fig.5. Still I am not convinced of the correctness of the execution of the Raman experiment. Authors referred to following papers claiming, that the Si peak in Raman spectra doesn’t have to be present. But: 1. In Nano. Res. Lett. 12 292 2017 they do refer to the Si peak (fig.2b) 2. In 10.1039/c4cp03582h authors didn’t show at all the Raman shift range, where the Si peak should appear. 3. I measured myself the rhodamine in top of SiO2 and I could see the Si peak. Still I am not convinced of the correctness of the execution of the Raman experiment. Quality of the inset in the fig2a is unacceptable,
=> It is authors' suggestion about what happens that the SiO2 (the reviewer used) can be many phases depending on many situations, that sometimes it has apparent Si-Si or some other peak of Si or sometimes it has totally Si-O phase. There are tons of different kinds of
SiO2. And more, our sample was silica (not silicon)-titania layer. It can be different from SiO2 which had Rhodamine in chemical structure.
This issue is again redundant. Authors provided already many publications not having the peak, which the reviewer have insisted on.
Even in the first revision, authors commented on the possibility of low sensitivity of our measurement system. However, there are many reports, that did not have the peak.